# Utilization of family planning and associated factors among women with disabilities in ethiopia: A systematic review and meta-analysis

**Tesfanesh Lemma Demisse**[1]*, **Mulualem Silesh**[1], **Birhan Tsegaw Taye**[1], **Tebabere Moltot**[1], **Moges Sisay Chekole**[1], **Maritu Ayalew**[2]

**1** Department of Midwifery, Asrat Woldeyes Health Science Campus, Debre Berhan University, Debre Berhan, Ethiopia, **2** Department of Midwifery, Teda Health Science College, Gondar, Ethiopia

* tesfitimnt@gmail.com

## Abstract

### Background

Persons with disabilities have a right to make their own choices about their bodies, health, and lives, especially regarding their sexual and reproductive health. But they may experience more challenges than women without disabilities in having their reproductive health needs met. So there is an urgent need to scale up disability inclusion in all levels of the health system including family planning. Therefore, the objective of this study was to estimate the pooled prevalence of Family Planning Utilization and Associated Factors among Women with Disabilities in Ethiopia.

### Methodology

Studies were gathered from Pub Med/MEDLINE (681), Google Scholar (426), African Journal of Online (AJOL) (36), CINAHL (211), HINARI (191), Scopus (86), Science Direct (62), Excerpta Medica database (EMBA, SE) (113), DOAJ (38), Web of Science (26), Google (271), and other organization's websites (2) using a combination of search terms and Boolean operators. The modified Newcastle Ottawa Scale (NOS) for cross-sectional research was used by three authors to independently assess the quality of each study. For statistical analysis, STATATM Version 11 software was employed. For the meta-analysis, the random-effects (Der Simonian and Laird) technique was applied. The heterogeneity test was performed using I-squared ($I^2$) statistics. A one-out sensitivity analysis was performed.

### Result

A total of 7 articles with 2787 participants were included in this systematic review and meta-analysis. The pooled prevalence of family planning utilization among Women with Disabilities was 29.6% (95% CI: 22.3, 36.8); I2 = 94.6%). Women who were in marital union (p<0.001) and who had a discussion with their husbands (p = 0.007) were factors significantly associated with the utilization of family planning among women with disabilities.

**Data Availability Statement:** All relevant data are within the manuscript and its Supporting Information files.

**Funding:** The author(s) received no financial support for the research, authorship, and/or publication of this article.

**Competing interests:** The authors have declared that no competing interests exist.

**Abbreviations:** CI, Confidence Interval; EDHS, Ethiopian Demographic and Health Survey; FP, Family Planning; MDG, Millennium Development Goals; SNNPR, Southern Nations, Nationalities and Peoples Region; WHO, World Health Organization; WWDs, Women with Disabilities; USAID, U.S. Agency for International Developmen.

## Conclusion

The finding of this study showed that utilization of family planning among women with disability is relatively lower than the Ethiopian Demographic Health Survey 2019. Therefore, the discussions with the partner and their engagement in decisions to use family planning are critical to increase its use.

## 1. Introduction

A disability is defined as a condition or function judged to be significantly impaired relative to the usual standard of an individual or group. The term is used to refer to individual functioning, including physical impairment, sensory impairment, cognitive impairment, intellectual impairment mental illness, and various types of chronic disease [1]. According to the 2021 World Health Organization (WHO) report, Over 1 billion people in the world live with some form of disability [2]. Women account for the majority of the disabled population in the world; globally, one in five women lives with a disability compared to one in eight men. In low and middle-income countries, women are estimated to comprise up to three-quarters of persons with disabilities [3].

Family planning is "the ability of individuals and couples to anticipate and attain their desired number of children and the spacing and timing of their births [4]. It is the information, means, and methods that allow individuals to decide if and when to have children [5]. Access to safe, voluntary family planning is a human right and is critical in ensuring gender equality and women's empowerment, and it is a key factor in reducing poverty and improving livelihoods [5–7]. Persons with disabilities have a right to make their own choices about their bodies, health, and lives, especially regarding their sexual and reproductive health and rights and freedom from discrimination and violence [8]. They have the same sexual and reproductive health needs and rights as people without disabilities, but often they are not given information about reproductive and sexual health or adequate care [9,10].

Studies indicate that disabled women were at higher risk of experiencing sexual violence than nondisabled women [11–13], and experience more challenges than women without disabilities in having their reproductive health needs met. Women with Disabilities (WWDs) face multiple barriers to quality contraceptive care [14], whether through stigma, increased risk of violence or abuse, lack of access to care [15], prejudices, and discrimination from healthcare service providers [14,16] and receive poor quality services [2,17].

So there is an urgent need to scale up disability inclusion in all levels of the health system, particularly primary health care [2] like family planning [15]. According to a review conducted in Low-and Middle-Income countries, contraception use among WWDs ranged from 13% to 31.1%, with 24.3 percent of unmet needs [18]. This study aimed to determine the pooled prevalence and associated factors for FP service utilization among WWDs in Ethiopia.

## 2. Methods and materials

### 2.1.Study design and search strategy

The purpose of this systematic review and meta-analysis was to assess the pooled prevalence of family planning utilization and associated factors among women with disabilities in Ethiopia using the Preferred Reporting Items for Systematic Reviews and Meta-analyses (PRISMA)

statement guidelines [19,20]. We searched the PROSPERO database to see whether there were any comparable on-going systematic reviews and meta-analyses.

Three authors searched the following databases for relevant studies: From Pub Med/MEDLINE (681), Google Scholar (426), African Journal of Online (AJOL) (36), CINAHL (211), HINARI (191),Scopus (86), Science Direct (62),Excerpta Medica database (EMBA, SE) (113), DOAJ (38),Web of Science (26),Google (271), and other organization's websites (2) by using the full title (Utilization of family planning and associated factors among women with disability in Ethiopia) and using the following searching keywords or terms; "Family planning utilization", "use of family planning methods, "utilization of family planning services", "State of family planning", "Family Planning Service Utilization", "practice of family planning", "Providing family planning services", "Contraceptive utilization", "Influencing Factors", "associated factors", " factors influencing", "Disabled women, "disables", "Women with disability", "disability", "disabilities" and, "Ethiopia" In order to find any further missed studies, the reference lists of all included published and unpublished studies were reviewed. All fields and MeSH (Medical Subject Headings) terms with Boolean operators ("OR" and/or "AND") were used to search studies in the advanced PubMed search engine (**S1 File**).

## 2.2.Eligibility criteria

Both published and unpublished observational studies in English that report the prevalence and/or associated factors of family planning utilization among disabled women in Ethiopia were included [21–27]. This review also included studies done up until November 15, 2022. Studies with a different operational definition or outcome of interest, as well as those whose full texts were unavailable, were excluded [28–32].

## 2.3.Outcome measurement, study selection, and quality assessment

Family planning utilization was considered when the women with disability had ever used any modern method of contraceptive during her sexual life and/or she is current user [23,26,27]. After studies were searched from different international databases and organization websites, studies were screened by using the following criteria (duplication, relevancy, accessibility of full text, and outcomes of interest). Finally, the quality of each study was assessed using the standard quality assessment tool [Newcastle-Ottawa Scale (NOS)] [33] and was assessed by five authors (TL, BTT, TM, MSC, and MA) independently using the following components: selection, comparability, and outcome; which were graded by five stars, two stars, and three stars respectively. Any disagreements between the five authors during quality appraisal were resolved by another author (MS) through discussion and re-evaluation of selected studies. For analysis, only the primary studies with a medium score (fulfilling 50% of the quality evaluation criteria) and above were included [34] (**S2 File**).

**Operational definition.** **Women with disability**: - women having hearing, visual and physical impairments or limb defects [23,27].

**Family planning utilization**: - The women with disability had ever used any modern method of contraceptive during her sexual life and or she is current user [23,26,27].

**Total fertility rate (TFR)**: - is the measure of children a women would have over her life time if she were to follow current age-specific fertility rates [26].

**Data extraction process.** Three independent authors (TL, MS, and BTT) extracted the data using a data extraction format prepared in a Microsoft Excel 2010 spreadsheet. The following information was extracted: the first author's name, the year of study, the study area, the Region, the study design, the sample size, the sampling method, the prevalence of family

planning, and the associated factors with their odds ratios. Differences during data extraction were resolved through discussion and consensus by involving the fourth author (TM).

**Data synthesis and statistical analysis.** For analysis, STATA Version 11 software was used. Because high heterogeneity across studies was identified using inverse variance ($I^2$) statistics with its corresponding p-value ($I^2 = 94.6\%$, $p < 0.001$) [35–37], a random effects model was applied to determine the pooled prevalence of family planning utilization. Meta-regression and subgroup analysis were also used to identify the source of heterogeneity across studies by using year of study, and region. To check for publication bias, a funnel plot and Egger's test were used [38]. The statistical significance of publication bias was determined using a p-value less than 0.05 [39]. Texts, tables, and forest plots with effect and 95% CI measures were used to present the results.

## 3. Result

### 3.1. Study selection

Using various search strategies, a total of 2080 studies were retrieved from various international databases and Ethiopian university institutional repositories. All retrieved studies were screened using the Endnote 7 reference manager, and 1421 were removed due to duplication. Then, 652 studies were eliminated due to unrelated titles, abstracts, inaccessibility of full text, and differences in outcomes of interest. Finally, for the meta-analysis, seven studies that met the inclusion criteria were considered (**Fig 1**).

### 3.2. Characteristics of included studies

This systematic review and meta-analysis included seven primary studies. All of the studies included were cross-sectional, with sample sizes ranging from 162–701 [25,26]. Regarding the study's region, three articles were from the Amhara region [22,23,26], two from Addis Ababa [24,25], one from SNNP [21] and one from the Tigray [27] region with years of study ranging from 2013 to 2020. The highest prevalence of family planning utilization was reported by Abera S. (44.4%) [25], which was done in the Addis Ababa, while the lowest utilization was reported in the Amhara region by Beyene GA, et al. (15.8%) (23) (**Table 1**).

### 3.3. Prevalence of family planning utilization among women with disability

A total of 7 (five published and two unpublished) studies with 2787 women were included in this systematic review and meta-analysis to estimate the pooled prevalence of family planning utilization among women with disabilities in Ethiopia. Accordingly, the overall estimated pooled prevalence of family planning utilization among women with disabilities with a random effects model was 29.57% (95% CI: 22.30, 36.83) with a heterogeneity index ($I^2$) of 94.6% (p = 0.000) (**Fig 2**).

### 3.4. Publication bias and Heterogeneity

To check for publication bias, a funnel plot and Egger's test [37] were used. The statistical significance of publication bias was determined using a p-value less than 0.05 [38]. Accordingly, the funnel plot results revealed a symmetrical pattern, indicating that the included studies do not have a publishing bias (**Fig 3**). Furthermore, Egger's regression test is not significant, indicating that there was no publication bias in the studies (**Table 2**).

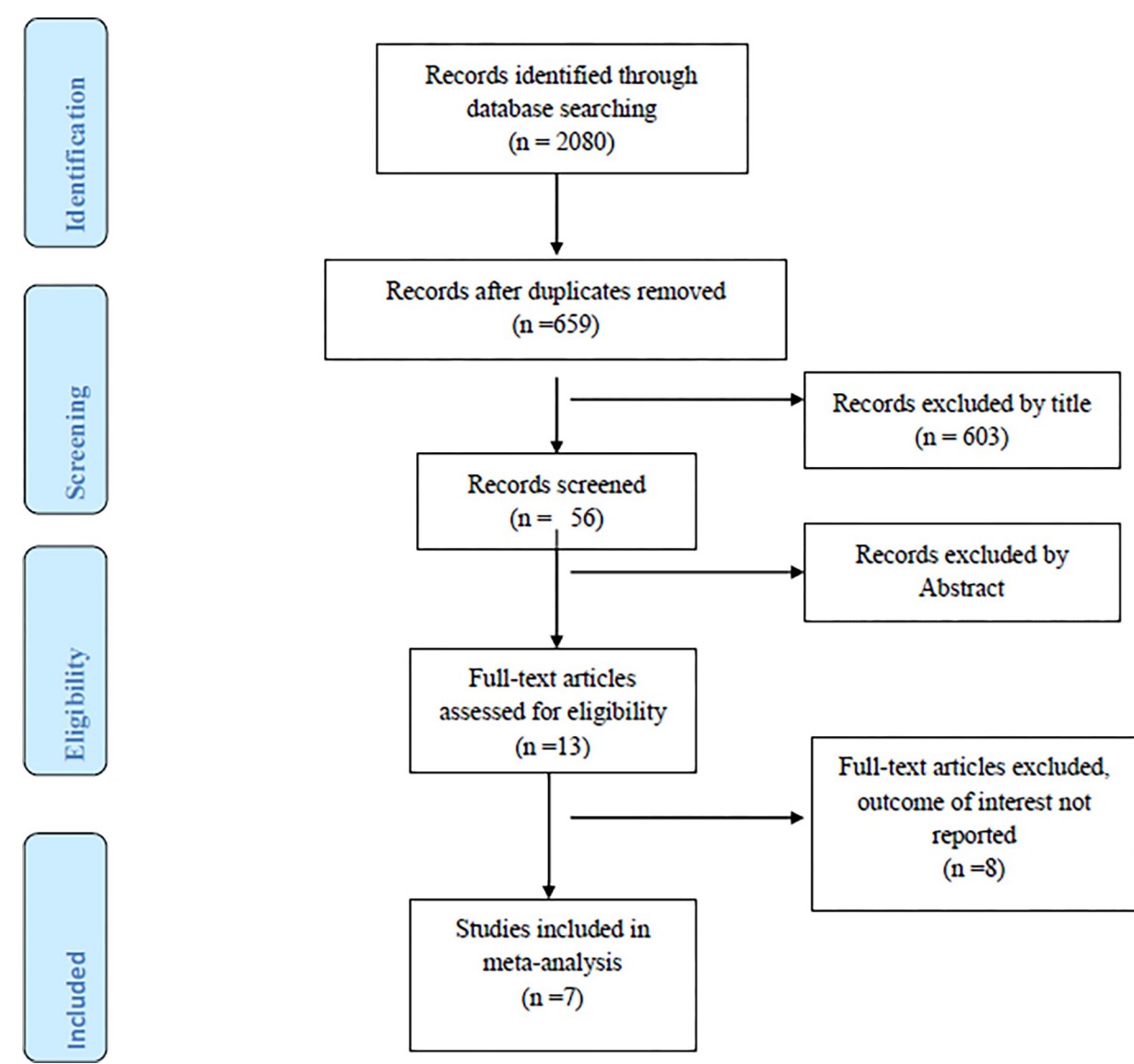

**Fig 1. PRISMA flow diagram of study selection.**

**Table 1. Characteristics of the included studies in the systematic review and meta-analysis.**

| Authors Name | Publication Year | Study setting | Region | Study design | sample | prevalence% (95% CI) |
|---|---|---|---|---|---|---|
| Yesgat YM, et al. | 2019 | Arba Minch | SNNPR | Cross Sectional | 398 | 33.7 (29.06, 38.34) |
| Mekonnen AG, et al. | 2019 | Debre Berhan | Amhara | Cross Sectional | 397 | 24.5 (20.27, 28.73) |
| Beyene GA, et al. | 2013 | Gondar | Amhara | Cross Sectional | 267 | 15.8 (11.42, 20.18) |
| Yimer AS, et al. | 2017 | Addis Ababa | Addis Ababa | Cross Sectional | 326 | 31.1 (26.07, 36.13) |
| Abera S (upulished) | 2016 | Addis Ababa | Addis Ababa | Cross Sectional | 701 | 44.4 (40.72, 48.08) |
| Tilahun A (upulished) | 2020 | Bahir Dar | Amhara | Cross Sectional | 162 | 30.2 (23.13, 37.27) |
| Tsegay K, et al. | 2013 | Mekelle | Tigray | Cross Sectional | 536 | 27.2 (23.43, 30.97) |

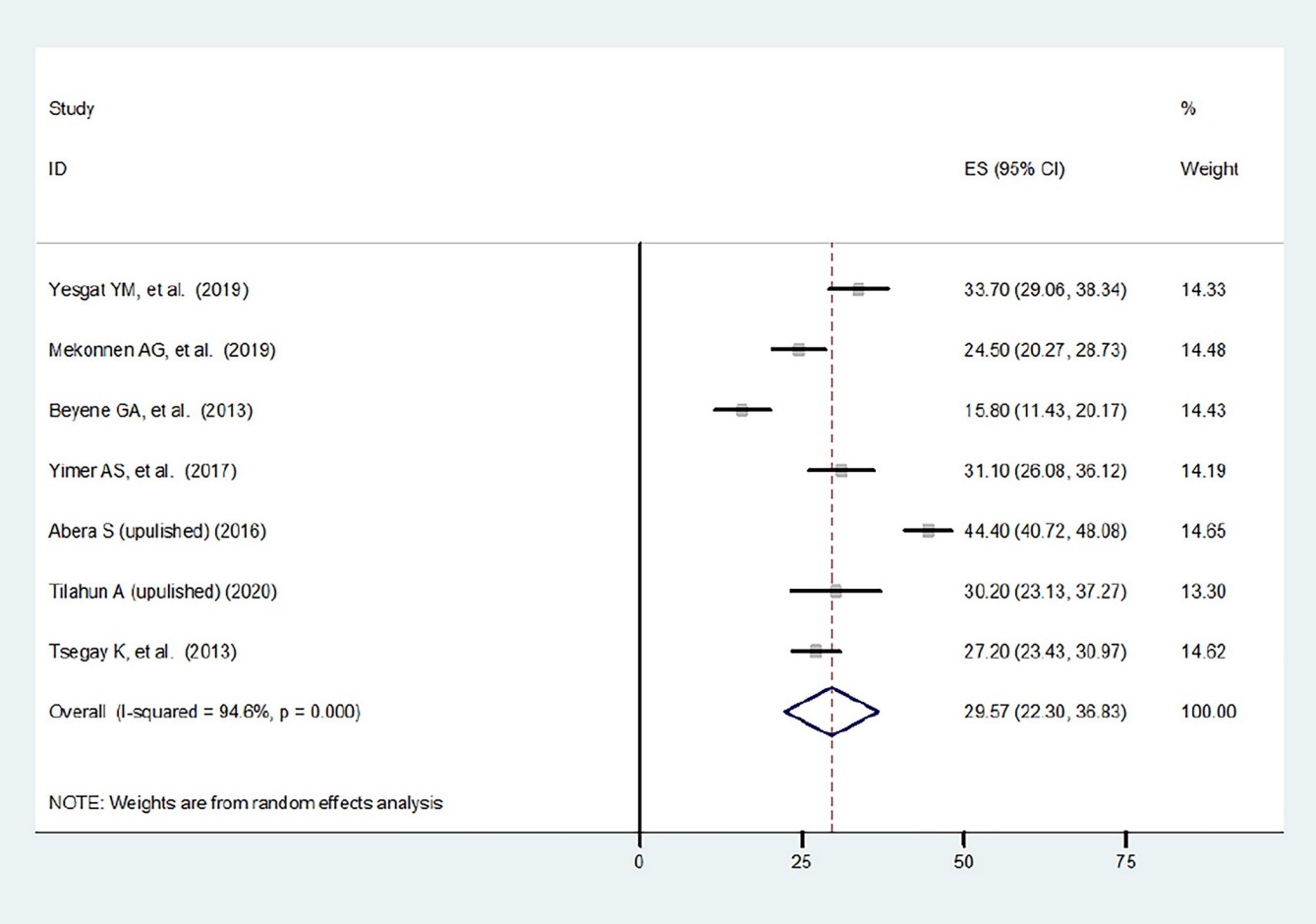

**Fig 2. Prevalence of family planning utilization among women with disability in Ethiopia.**

### 3.5. Subgroup analysis and sensitivity analysis

Significant heterogeneity was observed among included primary studies. To identify the source of heterogeneity, sub-group analysis was performed based on the region and year of study. As a result, the overall prevalence of family planning utilization was found to be high in Addis Ababa region studies [32.61, (26.39, 38.82)] (**Fig 4A**) and studies conducted after 2015 [32.61, (26.39, 38.82)] (**Fig 4B**).

A leave-one-out sensitivity analysis was done to check the effect of individual studies on the pooled estimate of family planning utilization. The result indicated removing a single study did not have a significant influence on pooled prevalence and the pooled prevalence of family planning utilization was observed low at 25.63% (3.93, 167.08%) and high at 31.78% (5.22, 193.66%) when Abera S. and Beyene GA, et al. were omitted respectively (**Table 3**).

### 3.6. Factors associated with family planning utilization among women with disability

Nine variables (age, marital status, discussion with partner, kowledge on FP methods, presence of FP providing nearby health facility, women's educational status, economic status, decision maker to use FP methods, and health facility keep the confidentiality & privacy) were extracted

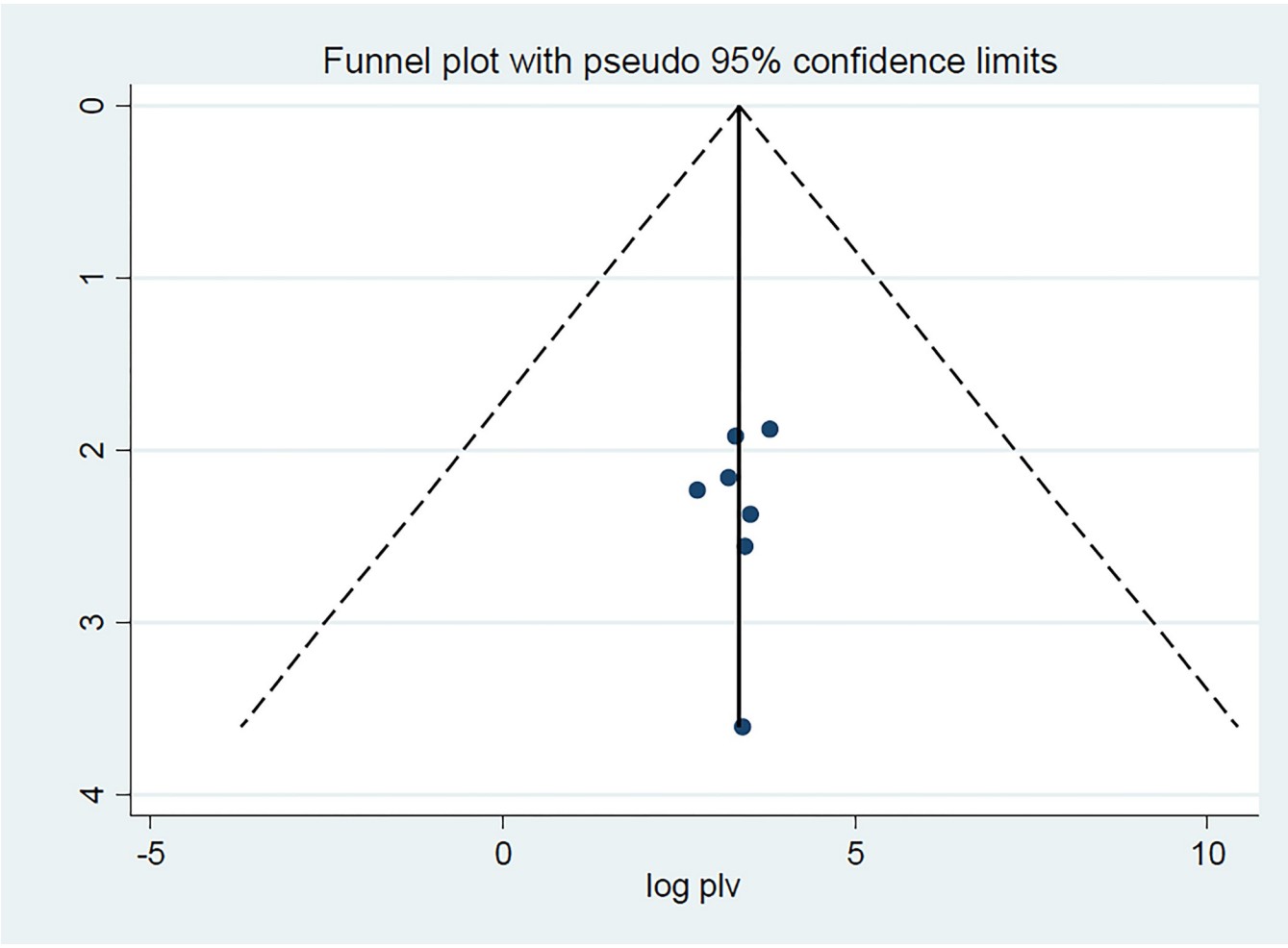

**Fig 3. Funnel plot to test the publication bias in 7 studies with 95% confidence limits.**

to identify factors associated with family planning utilization among women with disability. Of this marital status, discussion with a partner marital status, kowledge on FP methods, presence of FP providing nearby health facility, economic status, decision maker to use FP methods, and health facility keep the confidentiality & privacy were found to be significantly associated with family planning utilization among WWDs (**Table 4**). However, there was no statistically significant association between women's age, educational status and FP utilization.

The pooled effect of the marital status on FP utilization among women with disabilities was evaluated by using three primary studies (21,24,27) The result of this study revealed that marital status was significantly associated with FP utilization and the likelihood of utilizing FP was 8.6 times higher among those women who had in marital union than their counterparts [OR: 8.63; 95% CI (3.17, 23.46); P<0.001], with heterogeneity ($I^2$ = 91.5%, p-value <0.001) (**Fig 5**).

**Table 2. Egger's regression test to show publication bias in the studies.**

| Std.Eff | Coef. | Std. Err. | T | p> t | [95% Conf. Interval] |
|---------|-------|-----------|---|------|----------------------|
| Slope | 3.587349 | 0.7756355 | 4.63 | 0.006 | 1.593514,5.581183 |
| Bias | -0.1038034 | 0.3440247 | -0.30 | 0.775 | -0.988147,0.7805403 |

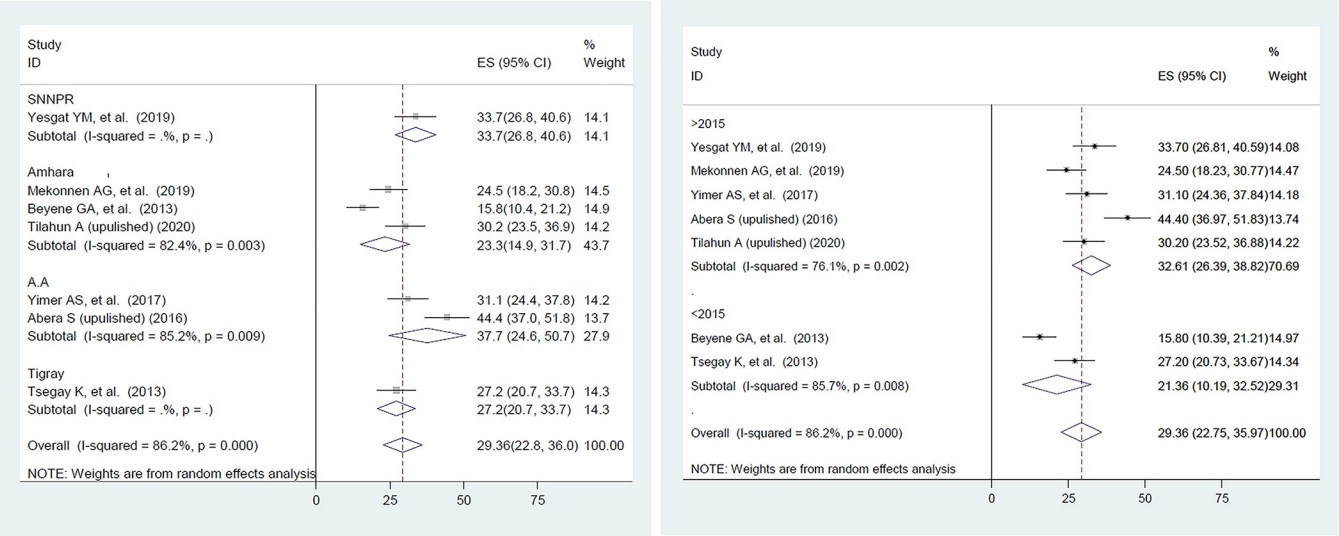

**Fig 4.** A. Subgroup analysis based on the region among women with disability in Ethiopia. B. Subgroup analysis based on year of study among women with disability in Ethiopia.

To determine the pooled effects of women's discussions with their partners on FP utilization, two studies were included [25,26]. The findings revealed that women's discussion with their partners was significantly associated with the use of FP; those who discussed with their partner were 6 times more likely to use FP than those who did not [OR: 6.08; 95% CI (1.64, 22.53); P = 0.007] with heterogeneity ($I^2$ = 84.2%, p-value = 0.012) **(Fig 6)**.

A total of two primary studies [21,22] were included to assess the association between knowledge on FP and utilization of FP among women with disabilities. Accordingly, women who had good knowledge were 1. 77 times more likely to utilize FP than those who had poor kowledge [OR = 1. 77, 95% CI = (1.22, 2.56); P = 0.003] with a mild type of heterogeneity between two variables ($I^2$ = 0.0%, p value = 0.383) **(Fig 7)**.

The finding of this study revealed that the economic status were significantly associated with FP utilization among women with disabilities [23] and the likelihood of utilizing FP service were 6.66 times higher in those mothers who had rich economic status [OR = 6.66, 95% (CI = 2.94,15.05); P <0.001] with heterogeneity index of ($I^2$ = 0.0%).This finding also revealed that decision maker to use family planning was significantly associated with the use of FP [26]; Disabled women whose decision to use family planning was made by their husband were 92% less likely to utilize family planning methods than women who made decisions by themselves [OR: 0.08; 95% CI (0.03, 0.21) P<0.001] with heterogeneity index ($I^2$ = 0%) **(Table 4)**.

**Table 3. Sensitivity analysis for utilization of family planning among women with disability in Ethiopia.**

| Study omitted | Year of study | Pooled estimate (%) | 95%CI |
|---|---|---|---|
| Yesgat YM, et al. | 2019 | 28.03 | 4.68,167.89 |
| Mekonnen AG, et al. | 2019 | 29.56 | 4.8, 182.05 |
| Beyene GA, et al. | 2013 | 31.79 | 5.22, 193.66 |
| Yimer AS, et al. | 2017 | 28.42 | 4.82, 167.02 |
| Abera S (upulished) | 2016 | 25.63 | 3.93, 167.08 |
| Tilahun A (upulished) | 2020 | 28.62 | 5.13, 159.65 |
| Tsegay K, et al. | 2013 | 29.09 | 4.51, 187.48 |

**Table 4. Factors associated with utilization of family planning among women with disability in Ethiopia.**

| Determinants | Comparisons | Number of studies | Sample size | OR(95% CI) | P- value | I² (%) | Heterogeneity test (P- value) |
|---|---|---|---|---|---|---|---|
| Marital Status | married Vs. unmarried | 3 | 1662 | 11.15 (4.82–11.98) * | 0.003 | 66.2 | <0.001 |
| Age | <25 Vs. ≥ 25 | 2 | 593 | 3.21 (1.01–10.18)* | <0.001 | 93.8 | 0.021 |
| Discussion with partner | Yes Vs. No | 2 | 649 | 2.49 (1.76–3.51)* | 0.301 | 6.5 | 0.012 |
| Kowledge o FP methods | Good Vs. Poor | 2 | 735 | 1.77 (1.22–2.56) * | 0.003 | 0 | 0.383 |
| Presence of FP providing nearby Health facility | Yes Vs. No | 2 | 698 | 1.49 (0.03–69.71) | 0.837 | 96.9 | <0.001 |
| Educational Status | Illiterate Vs. Literate | 1 | 267 | 1.81 (0.92–3.59) | 0.088 | 0 | <0.001 |
| Economic status | Poor Vs. Rich | 1 | 267 | 6.66 (2.94–15.05)* | <0.001 | 0 | - |
| Decision maker to use family planning | Husband Vs. Women | 1 | 162 | 0.08 (0.03–0.21) * | <0.001 | 0 | - |
| Health Workers keep the confidentiality & privacy | Yes Vs. No | 1 | 536 | 5.14 (3.13–8.43)* | <0.001 | 100 | - |

*Significant level <0.05.

Furthermore, the pooled effects of participants who trusted health workers as capable to keep their privacy were significantly associated with the utilization of family planning [27]. Accordingly, participants who trusted health workers as capable to keep their privacy was 5.14

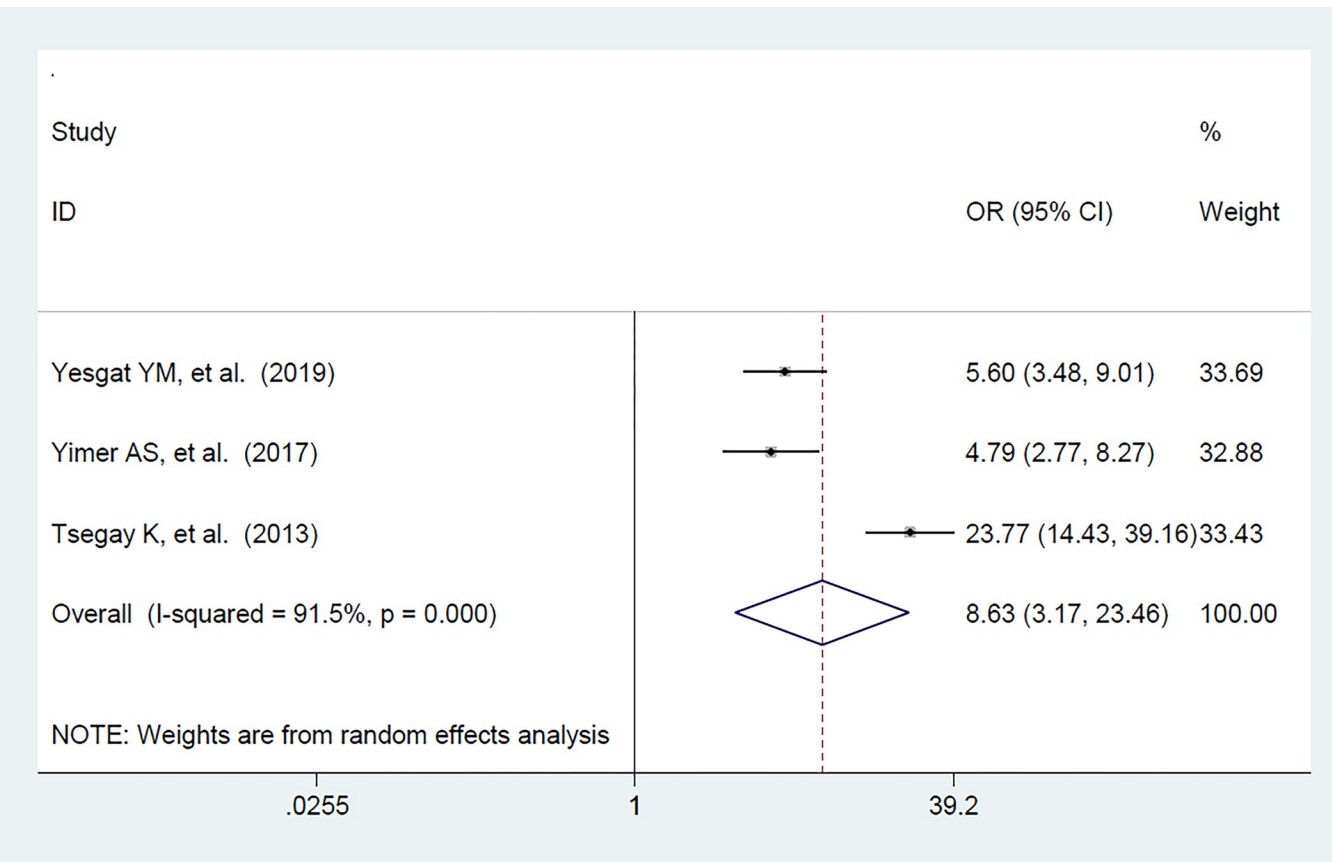

**Fig 5. The pooled effect of marital union on FP utilization among women with disabilities in Ethiopia.**

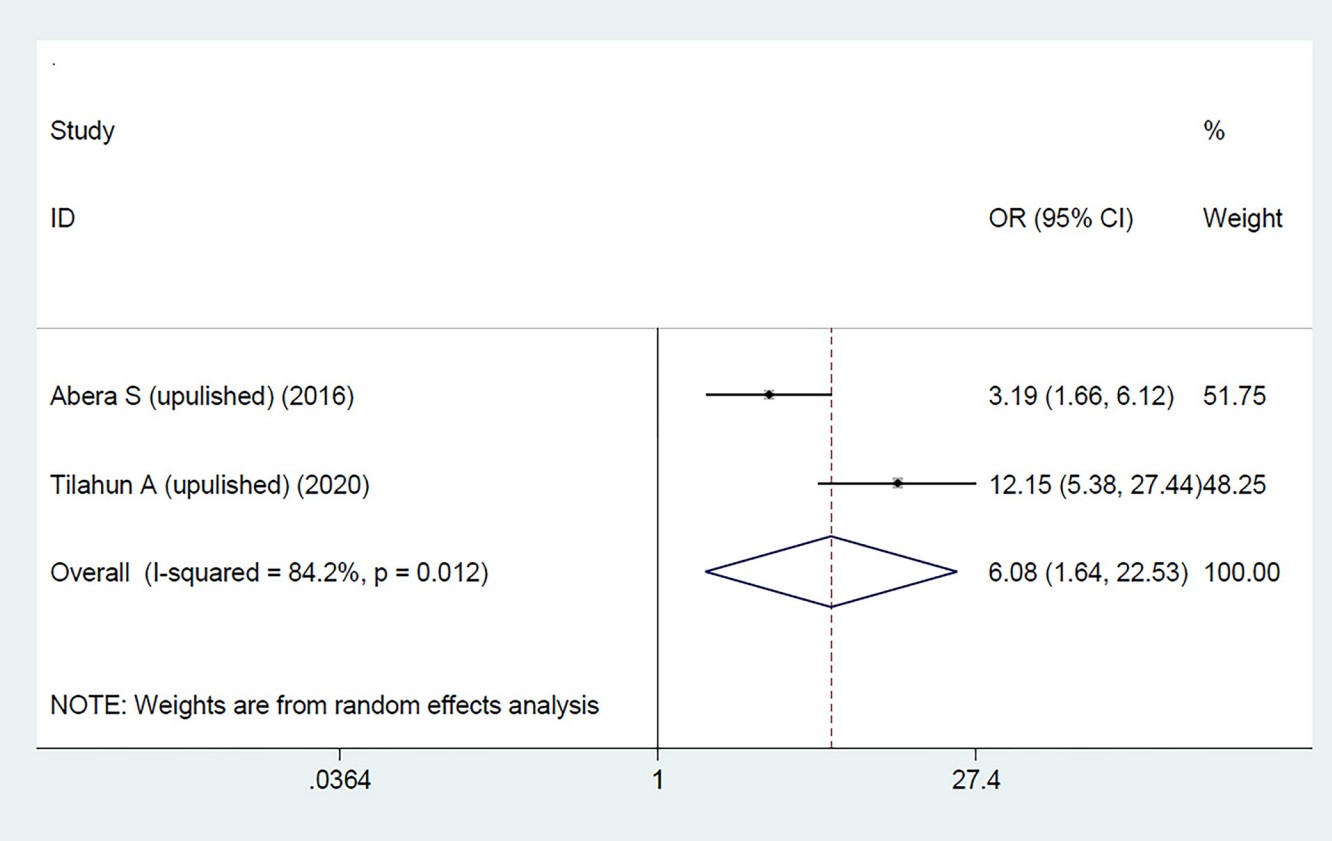

**Fig 6. The pooled effect of women's discussions with their partners on FP utilization among women with disabilities in Ethiopia.**

(3.13–8.43) times more likely to utilize family planning methods [OR = 3.73; 95% CI (3.13, 8.43) P<0.001] with heterogeneity index of 100% **(Table 4)**.

## 4. Discussion

The purpose of this systematic review and meta-analysis was to determine the pooled prevalence and associated factors of FP utilization among Ethiopian women with disabilities. To the best of our knowledge, this meta-analysis is the first of its kind in determining the national prevalence and significant factors of FP utilization among WWDs in Ethiopia, which will be used as input for policymakers, health care providers, and other stakeholders in developing evidence-based strategies to improve FP utilization among WWDs.

The overall pooled prevalence of family planning utilization among women with disabilities in Ethiopia was 29.57% (95% CI: 22.30, 36.83), which is lower than the Ethiopian Demographic and Health Survey (EDHS) report 2019 [40]. This could be because the EDHS reports family planning utilization among all women, but this review was limited to WWDs. Also, it is lower than a review conducted in Ethiopia [41,42]. The difference might be due to the difference in the study period of primary studies and the study population, since the two studies are conducted among adolescents and postpartum women who are more information about sexual and reproductive health from different sources.

However the result of this review was higher than the minimum rate of family planning utilization among WWDs, according to a review conducted by Horner-Johnson [43] and Beyene

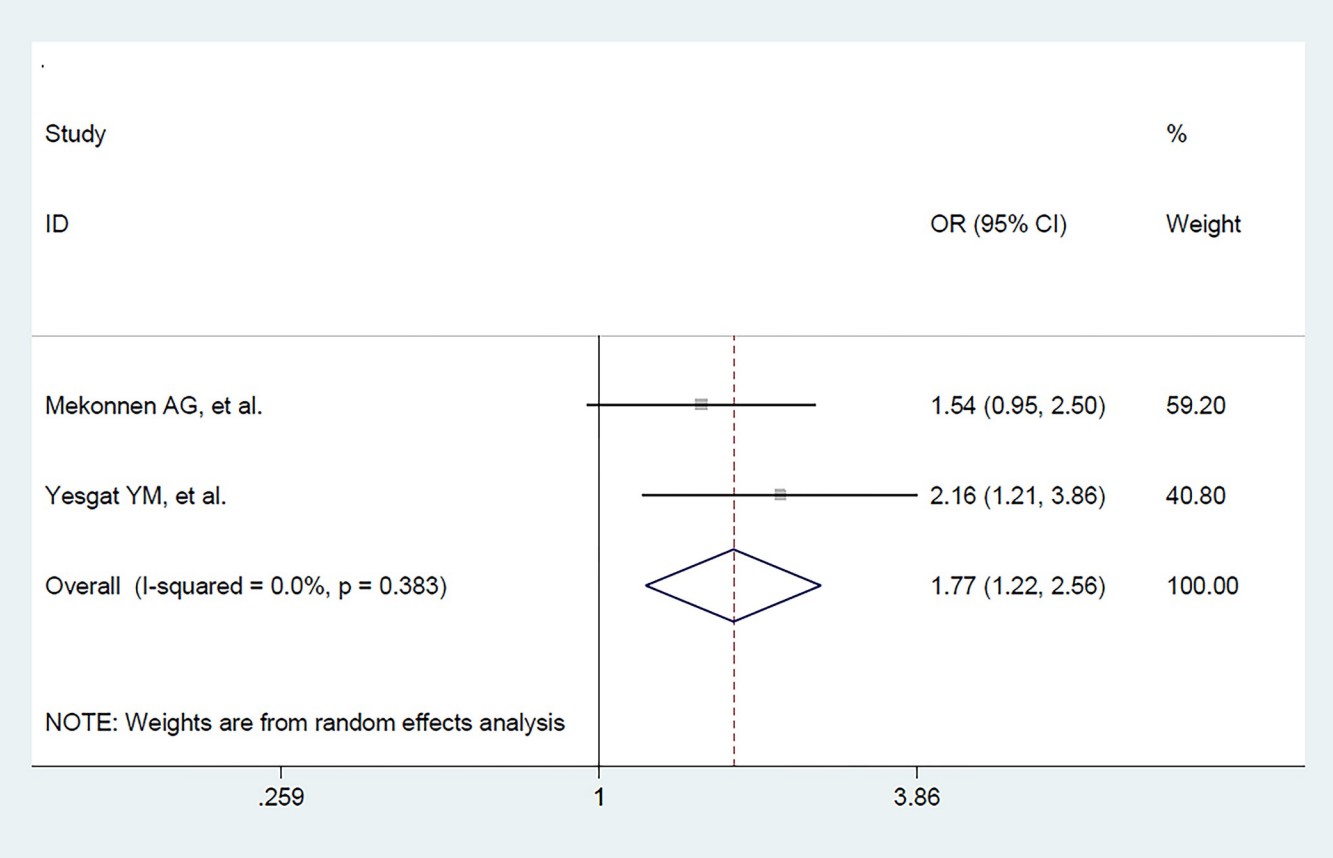

**Fig 7. The pooled effect of women's knowledge of FP on FP utilization among women with disabilities in Ethiopia.**

GA [18]. The result of this study could be a pooled estimate of different studies, whereas Horner-Johnson and Beyene GA report the minimum and maximum range of family planning utilisation among WWDs by reviewing literature from different settings, and the comparison in this study was based on the minimum rage from the report. In addition, it was higher than a review conducted by Y.F. Geda and T.M. Berhe [44]. The difference could be due to differences in study participants and contraceptive type, as this study was conducted among WWDs for the use of all types of contraceptives, whereas a study by Y.F. Geda and T.M. Berhe was conducted among postpartum women for the use of immediate postpartum intrauterine contraceptive devices (IUCD).

Based on the subgroup analysis result, the prevalence of family planning utilization was found to be high in Addis Ababa region studies [32.61, (26.39, 38.82)]. This could be attributed to the city's urbanization and women's increased awareness. Because Addis Ababa is the country's capital, disabled women may have more access to information on family planning. In Addis Ababa, there is also the Ethiopian Women with Disability Association, where many women participate and discuss their health, including family planning.

In addition, this review revealed that there is a high prevalence of family planning utilization among studies conducted after 2015 [32.61, (26.39, 38.82)]. The reason could be that after the Millennium Development Goals (MDG) were completed in 2015; The Ethiopian government developed a new plan called the Costed Implementation Plan for Family Planning, which includes various strategies to increase the contraceptive prevalence rate and decrease

total fertility rate by 2020, that may encourage more women to use contraception [45]. Furthermore, various information distribution technologies such as social media are increasingly being used to disseminate information regarding family planning options. This may make it easier for women to learn about the advantages of family planning, where to get it, and how to use it.

In this review, the odds of utilizing FP were 8.6 times higher among women who had been in a marital union than their counterparts. This is in line with a review conducted in sub-Saharan Africa [46], Ethiopia [47] and a report from USAID [48]. This is because family planning can increase partners' involvement in decisions about whether and when to have children while also assisting them in avoiding unintended pregnancy. It also provides her with enough time and opportunity to love and care for her husband and children. As a result, partner involvement in family planning decisions is critical, and a married woman may be encouraged to use contraception by her partner.

The findings of this study also revealed that women who discussed with their partner were 6 times more likely to use FP than those who did not. This is consistent with the review conducted in Sub-Saharan Africa [46,49]. This may be due to the fact that couples who talk about FP issues are more likely to jointly decide on the type of contraceptive method to use, the number of children to have, and the spacing between the children. As a result, they are more likely to use the service. Also Male participation in contraceptive use increases women's uptake, according to a report from a review protocol by Anbesu and his colleagues [50]. In addition male involvement plays a role in the use of reproductive and maternal health services, and any factor that influences the partner's attitude towards these services will have an effect on women's use, either positively or negatively. So talking with a partner may give them the impression that she values their influence in her life, giving the woman more freedom to make decisions.

This study found that disabled women who had good knowledge were significantly associated with utilization of family planning. This finding was supported by a review finding from Ethiopia [42,44] and around the globe [49]. This might be attributed to an in-depth knowledge of family planning methods, which can improve women's understanding and awareness of the importance and side effects of various contraceptive methods, allowing them to make an informed decision on the method to be used and, as a result, increase their usage of this service.

In this meta-analysis, economic status of the women was positively associated with FP utilization. Women who had rich economic status were more likely to utilize FP compared to their counterparts. This is consistent with a studies conducted by Mekonnen AG et.al [42]. This could be because women with higher economic status are more likely to be exposed to information on family planning due to their health seeking behavior for various reasons and their use of various information-gathering mass media such as the internet. The information may have influenced their use by helping them comprehend the purpose and importance of family planning methods.

In addition these reviews showed that a woman's whose decision to use family planning was made by their husband were less likely to utilize family planning methods than women who made decisions by themselves. This finding was consistent with a study finding from different settings [51,52]. This might be due to the fact that decision-making autonomy on contraceptive use influences their utilization. Women's independent decisions on reproductive health issues like FP are crucial and increase women's access to health information and utilization. So less autonomy in the decision regarding contraceptive use due to male dominance at the household level may affect their utilization.

Accordingly, this review revealed that participants who trusted health workers as capable to keep their privacy was more likely to utilize family planning which is supported by a review

conducted by Brittain et al. [53]. A possible explanation is that maintaining confidentiality and ensuring privacy are crucial for effective, sensitive management of potentially stigmatizing health conditions and improved quality care, including getting sexual and reproductive health services [54,55] So, if the woman believes the health care practitioner will respect her privacy and confidentiality, she is more likely to use the service.

There are limitations to this study. The absence of studies from some Ethiopian regions in this study makes it challenging to extrapolate the results to the national level. Second, the results should be interpreted cautiously due to the significant heterogeneity among the studies. Moreover, only observational study articles published in English were considered. Finally, we found it challenging to compare our results because there were few systematic reviews and meta-analyses conducted at the national, regional, and international levels.

## 5. Conclusion

According to the findings of this study, only one-third of disabled women in Ethiopia utilize family planning. Being married and discussing with a partner were significantly associated with family planning utilization among WWDs. Therefore, the discussions with the partner and their engagement in decisions to use family planning are critical to increase its use. It also important to increase access to quality contraceptive care and improve negative clinician attitudes and awareness of disabled women to increase utilization. In addition, it is critical to pay more attention to the reproductive health care needs of women with disabilities to improve health care equity.

## Supporting information

**S1 Checklist. PRISMA 2009 checklist.**
(DOCX)

**S1 File. Advanced PubMed search engine.**
(TXT)

**S2 File. Quality assessment of primary studies.**
(PDF)

## Author Contributions

**Conceptualization:** Tesfanesh Lemma Demisse.

**Data curation:** Tesfanesh Lemma Demisse, Mulualem Silesh, Birhan Tsegaw Taye, Tebabere Moltot, Moges Sisay Chekole, Maritu Ayalew.

**Formal analysis:** Tesfanesh Lemma Demisse, Mulualem Silesh.

**Investigation:** Tesfanesh Lemma Demisse.

**Methodology:** Tesfanesh Lemma Demisse, Mulualem Silesh, Birhan Tsegaw Taye, Tebabere Moltot, Moges Sisay Chekole.

**Resources:** Moges Sisay Chekole, Maritu Ayalew.

**Software:** Tesfanesh Lemma Demisse, Mulualem Silesh, Birhan Tsegaw Taye, Tebabere Moltot, Moges Sisay Chekole, Maritu Ayalew.

**Supervision:** Tesfanesh Lemma Demisse, Mulualem Silesh, Birhan Tsegaw Taye, Tebabere Moltot, Moges Sisay Chekole, Maritu Ayalew.

**Validation:** Tesfanesh Lemma Demisse, Mulualem Silesh, Birhan Tsegaw Taye, Tebabere Moltot, Moges Sisay Chekole.

**Visualization:** Tesfanesh Lemma Demisse, Mulualem Silesh, Birhan Tsegaw Taye, Tebabere Moltot.

**Writing – original draft:** Tesfanesh Lemma Demisse.

**Writing – review & editing:** Tesfanesh Lemma Demisse, Mulualem Silesh, Birhan Tsegaw Taye, Tebabere Moltot, Moges Sisay Chekole, Maritu Ayalew.

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
