## [Decision Letter · Decision Letter 0]

16 May 2023

PONE-D-22-35356Utilization of Family Planning and Associated Factors among Women with Disabilities in Ethiopia: a systematic review and meta-analysisPLOS ONE

Dear Dr. Lemma,

Thank you for submitting your manuscript to PLOS ONE. After careful consideration, we feel that it has merit but does not fully meet PLOS ONE’s publication criteria as it currently stands. Therefore, we invite you to submit a revised version of the manuscript that addresses the points raised during the review process.

ACADEMIC EDITOR: Methods- You said the PROSPERO registration is ongoing, what is the importance of registering it after you have published the review?- It is better of specify the number of articles you have gotten from each data bases (Pub Med/MEDLINE=? Google Scholar=? African Journal of Online (AJOL)=? CINAHL=?, HINARI=?, Scopus=?, ScienceDirect, Excerpta Medica database (EMBA, SE), DOAJ, Web of Science, Google, and other organization's websites=?)- What is the final mesh term used in each searching engines?- Is publication year or study period better indicating the time period of the articles?Result- What does it mean p = 0.000?- you have checked three independent variables to assess their association with the outcome variable, some single articles revealed more than three significant variables?- the discussion is shallow, it needs to entertain more perspectives

We look forward to receiving your revised manuscript.

Kind regards,

PLOS ONE

Reviewers' comments:

Reviewer's Responses to Questions

**Comments to the Author**

1. Is the manuscript technically sound, and do the data support the conclusions?

Reviewer #1: Yes

Reviewer #2: Partly

2. Has the statistical analysis been performed appropriately and rigorously? 

Reviewer #1: N/A

Reviewer #2: Yes

3. Have the authors made all data underlying the findings in their manuscript fully available?

Reviewer #1: Yes

Reviewer #2: Yes

4. Is the manuscript presented in an intelligible fashion and written in standard English?

Reviewer #1: Yes

Reviewer #2: No

5. Review Comments to the Author

Reviewer #1: the matanalysis is very impotant as diasbilitis is a topic very poorly reported in current literture. this study underlines in ethiopoiaa those aspects in women with disabilities encoiunterin pregancy

Reviewer #2: Review Report

Title: Utilization of Family Planning and Associated Factors among Women with Disabilities in Ethiopia: a systematic review and meta-analysis.

Manuscript Number: PONE-D-22-35356.

Comments

a. Acknowledge for addressing disadvantaged segment of the community.

b. Scope: The scope of the issue and the outcome variables needs re-operationalization. The authors failed to include fertility.

c. Methods: Whether retracted articles are used in the analysis were not mentioned,

d. Illegibility: Not included all observational studies.

e. Whether all studies defined disability in the similar way: Not explained.

f. Data analysis and Presentation: Inadequate.

g. Authorship: inconsistent in the main document and in the methods section.

h. Language and statistic: Need major revision.

Regards,

6. PLOS authors have the option to publish the peer review history of their article (what does this mean?). If published, this will include your full peer review and any attached files.

Reviewer #1: **Yes: **Erich Cosmi

Reviewer #2: No

---

## [Author Response · Author response to Decision Letter 0]

23 Jun 2023

The authors would like to thank the editorial team and team of reviewers for constructive and valuable comments. The authors are very happy to submit the revised version of the manuscript entitled “Utilization of Family Planning and Associated Factors among Women with Disabilities in Ethiopia: a systematic review and meta-analysis” for its publication in your Journal. The comments of the editors and the reviewers were highly insightful and enabled us to greatly improve the quality of our manuscript. In this revised manuscript we made substantial changes to address your concerns in a point-by-point response. We are very keen to incorporate further comments, if any, for the betterment of the final manuscript.

Response to the Editor 

Methods

1. You said the PROSPERO registration is on-going, what is the importance of registering it after you have published the review?

Answer –Thank you for your comment; sorry for the inconvenience, and it is corrected in the revised manuscript.

2. It is better of specify the number of articles you have gotten from each data bases (Pub Med/MEDLINE=? Google Scholar=? African Journal of Online (AJOL)=? CINAHL=?, HINARI=?, Scopus=?,Science Direct, Excerpta Medica database (EMBA, SE), DOAJ, Web of Science, Google, and other organization's websites=?)

Answer – Thank you for your question. Here are the number of articles from each data base Pub Med/MEDLINE=681 Google Scholar=426 African Journal of Online (AJOL) =36 CINAHL=211, HINARI=191, Scopus=86, Science Direct=62, Excerpta Medica database (EMBA, SE) = 113, DOAJ= 38, Web of Science=26, Google= 271 and organization's websites=2)

 

3. What is the final mesh term used in each searching engines?

Answer –Thank you for your questions; you can see additional file 1 to understand the whole search engine by one of the most usable data base PubMed. 

Here are some of the mesh terms that are used in searching engines. 

Word Mesh terms 

Utilization of Family Planning Family planning utilization, use of family planning methods, utilization of family planning services, State of family planning, Family Planning Service Utilization, practice of family planning, Providing family planning services, Contraceptive utilization

Factors associated with family planning Influencing Factors, associated factors, factors, factors influencing, 

Women with disabilities Disabled women, disables , Women with disability, disability, disabilities, 

4. Is publication year or study period better indicating the time period of the articles?

Answer –Thank you for your questions; the study period is better indicating the time period of the articles and publication year also used to determine how old the information can be.

Result

1. What does it mean p = 0.000?

Answer –Thank you for your comment; it is an editorial problem and corrected in the revised manuscript 

2. You have checked three independent variables to assess their association with the outcome variable, some single articles revealed more than three significant variable.

Answer – Thank you for your inquiry; you are right that some single articles revealed more than three significant variable. But the only factor identified as a significant factor in the two and above primary studies was included in this review and meta-analysis. That is the reason for only three independent variables were assessed for their association. 

3. The discussion is shallow, it needs to entertain more perspectives

Answer –Thank you for your comment; it is accepted and corrected in the revised manuscript.

Response to the reviewer

Reviewer #1: the Meta analysis is very important as a disability is a topic very poorly reported in current literature. This study underlines in Ethiopia those aspects in women with disabilities encountering pregnancy.

Answer –Thank you for your insightful idea. 

Reviewer #2: 

A. Acknowledge for addressing disadvantaged segment of the community.

Answer –Thank you for your idea. 

B. Scope: The scope of the issue and the outcome variables needs re-operationalization. The authors failed to include fertility.

Answer – Thank you for your comment; it is accepted and corrected on page 5 line 22-28 of the revised manuscript.

C. Methods: Whether retracted articles are used in the analysis were not mentioned,

Answer – Thank you for your comment; as mentioned in the Prisma diagram(figure 1), 13 articles were retracted to assess their eligibility, but only seven studies were included in the analysis and the other six articles were excluded because of the outcome of interest is not reported. It is also described on page 6, line 19-24 of the revised manuscript. 

D. Illegibility: Not included all observational studies.

Answer – Thank you for your inquiry; First the authors plan to include all observational studies but all studies who fulfil the eligibility criteria in this review were crossectional, that’s why the authors don’t included all observational studies. 

E. Whether all studies defined disability in the similar way: Not explained

Answer – Thank you for your suggestion, majority of the studies included in this review defined people with disability as women having hearing, visual and physical impairments or limb defects (1–5). This is described in the revised manuscript on page 5, line 23-24.

F. Data analysis and Presentation: Inadequate.

Answer – Thank you for your comment. The authors carried out all of the necessary analysis to show the findings of the meta-analysis, such as a random effects model that was used to determine the pooled prevalence of family planning utilisation, as shown in Figure 2. Because there is heterogeneity across studies, meta-regression and subgroup analysis were used to identify the source of heterogeneity using the sample size, year of study, and region, as shown in Figures 4A and 4B. A funnel plot and Egger's test were used to check for publication bias, as shown in Figure 3 and Table 2. A leave-one-out sensitivity analysis was also performed to see how individual studies affected the pooled estimate of family planning utilisation. The results showed that removing a single study had no significant effect on pooled prevalence, as shown in Table 3.

G. Authorship: inconsistent in the main document and in the methods section.

Answer – Thank you for your comment; it is accepted and corrected in the revised manuscript.

H. Language and statistic: Need major revision.

Answer – Thank you for your feedback; it has been accepted, and the paper has been revised by language experts and online writing tools like Grammarly and QuillBot. As previously stated, this study performed all of the necessary analysis in order to show the results of the meta-analysis. 

Reference 

1. Beyene GA, Munea AM, Fekadu GA. Modern contraceptive use and associated factors among women with disabilities in gondar city, amhara region, north west ethiopia: A cross sectional study. Afr J Reprod Health. 2019;23(2):101–9. 

2. Kellali T, Hadush G FH. Modern Contraceptive Methods Utilization and Associated Factors among Women with Disabilities. Int J Pharm Biol Sci Fundam [Internet]. 2017;13(01)(01):1–8. Available from: www.ijpbsf.com

3. Mekonnen AG, Bayleyegn AD, Aynalem YA, Adane TD, Muluneh MA, Asefa M. Level of knowledge, attitude, and practice of family planning and associated factors among disabled persons, north-shewa zone, Amhara regional state, Ethiopia. Contracept Reprod Med. 2020;5(1):1–7. 

4. Mesfin Yesgat Y, Gebremeskel F, Estifanous W, Gizachew Y, Jemal S, Atnafu N, et al. 

Utilization of Family Planning Methods and Associated Factors Among Reproductive-Age Women with Disability in Arba Minch Town, Southern Ethiopia

. Open Access J Contracept. 2020;Volume 11:25–32. 

5. Abera S. The Assessment of Determinants of family planning use and unmet need among women of reproductive age group with disabilities in Addis Ababa. 2016;(November).

---

## [Editor Report · Decision Letter 1]

28 Jun 2023

PONE-D-22-35356R1Utilization of Family Planning and Associated Factors among Women with Disabilities in Ethiopia: a systematic review and meta-analysisPLOS ONE

Dear Dr. Lemma,

Thank you for submitting your manuscript to PLOS ONE. After careful consideration, we feel that it has merit but does not fully meet PLOS ONE’s publication criteria as it currently stands. Therefore, we invite you to submit a revised version of the manuscript that addresses the points raised during the review process.

ACADEMIC EDITOR: I have attached the comments.==============================

We look forward to receiving your revised manuscript.

Kind regards,

Lebeza Alemu Tenaw

Academic Editor

PLOS ONE

Additional Editor Comments :

I have attached the additional comments that should be considered in the next revision.

---

## [Author Response · Author response to Decision Letter 1]

17 Jul 2023

The authors would like to thank the editorial team for constructive and valuable comments. The authors are very happy to submit the revised version of the manuscript entitled “Utilization of Family Planning and Associated Factors among Women with Disabilities in Ethiopia: a systematic review and meta-analysis” for its publication in your Journal. The comments were highly insightful and enabled us to greatly improve the quality of our manuscript. In this revised manuscript we made substantial changes to address your concerns in a point-by-point response. We are very keen to incorporate further comments, if any, for the betterment of the final manuscript.

1. Comment [A1]: As I have said in the previous comment the identified articles from each data base is needed…..PubMed=-----?, CINARI=----? For each databases

Answer –Thank you for your comment; it is accepted and corrected in the revised manuscript.

2. Comment [A2]: I haven’t seen the logical evidences to exclude case control study in this review.

Answer – Thank you for your inquiry; First the authors plan to include all observational studies but all studies who fulfil the eligibility criteria in this review were crossectional, that’s why the authors don’t included other observational studies like case control studies

3. Comment [A3]: What is your standard to say this finding is low?

Answer – According to the Ethiopian Demographic Health Survey 2019, this finding is low.

4. Comment [A4]: Have you got in your finding which showed health care equity and quality were the main determinant factors for low FP service utilization?

Answer – Thank you for your comment; it is accepted and corrected in the revised manuscript. 

5. Comment [A5 & A6]: This study aimed to determine the pooled prevalence and associated factors for FP service utilization among WWDs in Ethiopia; better to state it at the end of the last paragraph of the introduction.

Answer – Thank you for your comment; it is accepted and corrected in the revised manuscript. 

6. Comment [A7]: Thank you for your comment; it is accepted and corrected in the revised manuscript.

7. Comment [A8]: Which articles were included or excluded based on this reason.

Answer – Thank you for your inquiry; the included and excluded studies are cited in the revised manuscript. 

8. Comment [A9]: Thank you for your comment; it is accepted and corrected in the revised manuscript.

9. Comment [A10]: Is there any logical evidence to exclude a single variable which showed significant association in the single study?

Answer– Even though there is no clear evidence to exclude a single variable that showed a significant association in the single study, meta-regression is the aggregate result of two or more studies. So to get the pooled results of different studies, the authors preferred to extract factors that were reported in two or more studies. 

10. Comment [A11]: Better to state all variables even they don’t have significant association

Answer– Thank you for your comment; as we mentioned earlier in this study the variable was extracted as factors, if it is significant associations in two or more primary papers. So, in this study only three variables were extracted as a factor and among them two of them are significantly associated with the dependent variable which is seen in Table 4. 

11. Comment [A13]: You have used very limited literatures for discussion.

Answer – Thank you for your comment; it is accepted and corrected in the revised manuscript. 

12. Comment [A14]: Thank you for your comment; it is accepted and corrected in the revised manuscript.

---

## [Editor Report · Decision Letter 2]

20 Jul 2023

PONE-D-22-35356R2Utilization of Family Planning and Associated Factors among Women with Disabilities in Ethiopia: a systematic review and meta-analysisPLOS ONE

Dear Dr. Lemma,

Thank you for submitting your manuscript to PLOS ONE. After careful consideration, we feel that it has merit but does not fully meet PLOS ONE’s publication criteria as it currently stands. Therefore, we invite you to submit a revised version of the manuscript that addresses the points raised during the review process.

We look forward to receiving your revised manuscript.

Kind regards,

Additional Editor Comments :

- The reason to exclude case control studies not convincing.

- Excluding variables which are significant in the single study is not advisable; we can take odds ration even though it may be not significant.

---

## [Author Response · Author response to Decision Letter 2]

15 Aug 2023

Response to the Editor 

1. The reason to exclude case control studies not convincing.

Answer: Thank you for your comment, as the authors mentioned in the manuscript on page 5 lines 2-4, both published and unpublished observational studies in English that report the prevalence and/or associated factors of family planning utilization among disabled women in Ethiopia were considered. But all studies which fulfill the inclusion criteria are crossectional, that’s why all the included studies were crossectional. 

2. Excluding variables which are significant in the single study is not advisable; we can take odds ration even though it may be not significant. 

Answer: Thank you for your comment, its accepted and corrected as your kind recommendation in the revised manuscript on pages 9 and 10.

---

## [Editor Report · Decision Letter 3]

24 Aug 2023

Utilization of Family Planning and Associated Factors among Women with Disabilities in Ethiopia: a systematic review and meta-analysis

PONE-D-22-35356R3

Dear Dr.Tesfanesh,

We’re pleased to inform you that your manuscript has been judged scientifically suitable for publication and will be formally accepted for publication once it meets all outstanding technical requirements.

Kind regards,

Academic Editor

PLOS ONE

---

## [Editor Report · Acceptance letter]

1 Sep 2023

PONE-D-22-35356R3 

Utilization of Family Planning and Associated Factors among Women with Disabilities in Ethiopia: a systematic review and meta-analysis 

Dear Dr. Demisse:

I'm pleased to inform you that your manuscript has been deemed suitable for publication in PLOS ONE. Congratulations! Your manuscript is now with our production department. 

Kind regards, 

on behalf of

Mr. Lebeza Alemu Tenaw 

Academic Editor

PLOS ONE